# Central Alteration in Peripheral Neuropathy of Trembler-J Mice: Hippocampal pmp22 Expression and Behavioral Profile in Anxiety Tests

**DOI:** 10.3390/biom11040601

**Published:** 2021-04-19

**Authors:** Juan Pablo Damián, Lucia Vázquez Alberdi, Lucía Canclini, Gonzalo Rosso, Silvia Olivera Bravo, Mariana Martínez, Natalia Uriarte, Paul Ruiz, Miguel Calero, María Vittoria Di Tomaso, Alejandra Kun

**Affiliations:** 1Unidad de Bioquímica, Departamento de Biociencias Veterinarias, Facultad de Veterinaria, Universidad de la República, 11600 Montevideo, Uruguay; jpablodamian@gmail.com; 2Departamento de Proteínas y Ácidos Nucleicos, Instituto de Investigaciones Biológicas Clemente Estable, 11600 Montevideo, Uruguay; lvazquez@iibce.edu.uy (L.V.A.); marimartinezbarreiro@gmail.com (M.M.); 3Departamento de Genética, Instituto de Investigaciones Biológicas Clemente Estable, 11600 Montevideo, Uruguay; lcanclini@gmail.com (L.C.); marvi@iibce.edu.uy (M.V.D.T.); 4Max Planck Institute for the Science of Light, Max-Planck-Zentrum für Physik und Medizin, 91058 Erlangen, Germany; gonzalo.rosso@mpl.mpg.de; 5Institute of Physiology II, University of Münster, 48149 Münster, Germany; 6Neurobiología Celular y Molecular, Instituto de Investigaciones Biológicas Clemente Estable, 11600 Montevideo, Uruguay; solivera@iibce.edu.uy; 7Laboratorio de Neurociencias, Facultad de Ciencias, Universidad de la República, 11400 Montevideo, Uruguay; natiuria@fcien.edu.uy; 8Unidad de Biofísica, Departamento de Biociencias Veterinarias, Facultad de Veterinaria, Universidad de la República, 11600 Montevideo, Uruguay; paulruiz@fvet.edu.uy; 9Unidad de Encefalopatías Espongiformes, UFIEC, CIBERNED, CIEN Foundation, Queen Sofia Foundation Alzheimer Center, Instituto de Salud Carlos III, 28031 Madrid, Spain; mcalero@isciii.es; 10Sección Bioquímica, Facultad de Ciencias, Universidad de la República, 11400 Montevideo, Uruguay

**Keywords:** Charcot–Marie–Tooth, hippocampus, peripheral-myelin-protein-22, anxiety, Trembler-J, CA3 neurons

## Abstract

Charcot–Marie–Tooth (CMT) type 1 disease is the most common human hereditary demyelinating neuropathy. Mutations in pmp22 cause about 70% of all CMT1. Trembler-J (TrJ/+) mice are an animal model of CMT1E, having the same spontaneous pmp22 mutation that is found in humans. We compared the behavior profile of TrJ/+ and +/+ (wild-type) in open-field and elevated-plus-maze anxiety tests. In these tests, TrJ/+ showed an exclusive head shake movement, a lower frequency of rearing, but a greater frequency of grooming. In elevated-plus-maze, TrJ/+ defecate more frequently, performed fewer total entries, and have fewer entries to closed arms. These hippocampus-associated behaviors in TrJ/+ are consistent with increased anxiety levels. The expression of pmp22 and soluble PMP22 were evaluated in E17-hippocampal neurons and adult hippocampus by in situ hybridization and successive immunohistochemistry. Likewise, the expression of pmp22 was confirmed by RT-qPCR in the entire isolated hippocampi of both genotypes. Moreover, the presence of aggregated PMP22 was evidenced in unmasked granular hippocampal adult neurons and shows genotypic differences. We showed for the first time a behavior profile trait associated with anxiety and a differential expression of pmp22/PMP22 in hippocampal neurons of TrJ/+ and +/+ mice, demonstrating the involvement at the central level in an animal model of peripheral neuropathy (CMT1E).

## 1. Introduction

Charcot–Marie–Tooth disease type 1 (CMT1) is the most common hereditary demyelinating neuropathy [1,2,3,4,5]. In addition to diverse sensory, motor, and clinical symptoms, CMT1 is associated with distinct genetic patterns related to autosomal dominant or recessive inheritance as well as with sporadic cases. Alterations in the pmp22 gene are the most common cause of CMT1, along with hereditary neuropathy with liability to pressure palsy (HNPP) [6,7,8]. In addition to duplications (CMT1A) and deletions (HNPP), all pmp22 point mutations have recently been included as a subtype of CMT1, CMT1E.

PMP22 is a hydrophobic integral membrane glycoprotein highly expressed in Schwann cell myelin. It is composed of 160 amino acids, with a molecular mass of 22 kDa and a single N-glycosylation site [9,10,11,12]. The name peripheral myelin protein was suggested by Snipes et al. 1992, who reported that “PMP-22 is synthesized by Schwann cells and is a major component of Peripheral Nervous System (PNS), but not Central Nervous System (CNS) myelin”. Nevertheless, some experimental evidence has been reported, indicating the presence of pmp22 transcripts in RNA extracts from the entire brain [13,14,15]. Likewise, an in situ hybridization (ISH) study showed that pmp22 mRNA and protein are present only in the cranial and spinal nerve motoneurons [10]. On the other hand, familial CMT1A cases with concomitant lesions of the brain white matter were revealed by magnetic resonance imaging (MRI) [16]. The authors proposed an autoimmune mechanism for lesions of the brain white matter related to the production of autoantibodies against PMP22, thus suggesting the presence of the protein in the CNS. Later, the transcripts of the PMP22 in the CNS were demonstrated by Northern blot [17]. This study also reported the presence and localization of PMP22 protein in the myelin of peripheral nerves, ventral and dorsal roots, and in motor neurons and preganglionic sympathetic neurons in the spinal cord [17]. More recently, in a patient with another type of CMT (CMTX1), MRI studies showed abnormalities in the brain, specifically in the bilateral centrum semiovale and splenium of the corpus callosum [18].

Although some studies have reported CNS clinical manifestations in individuals with CMT, the data about animal models is scarce. Norreel et al. (2001) evaluated the behavioral profile in two transgenic mice models of CMT1A, reporting peripheral but not central motor deficits [19]. The authors reinforce the hypothesis based on Huxley et al. (1996), who did not find transgene expression in the CNS [20]. However, unlike CMT1A, related to the duplication of the 17p11.2–p12 region of pmp22, CTM1E is associated with point mutations that produce structural modification of the PMP22 protein [21]. One of these mutations, the substitution of proline for leucine (L16P), generates incorrect folding of PMP22, preventing its correct insertion into the cell membrane, which is associated with a loss of function [22,23] and becoming toxic for Schwann cells [12,24,25,26]. Therefore, there are clear differences in the pathogenesis of these two different subtypes of CMT (CMT1A vs. CMT1E), which correlate with different phenotypes. In this sense, Tremble-J mice represent a valuable animal model of CMT1E, sharing with humans beings [27] the same pmp22 spontaneous substitution (T1703C) given as L16P PMP22 final modification [6,7,8,22,26,28]. The heterozygous TrJ/+ mouse presents spastic paralysis, progressive limb weakness, and a generalized tremor, while the homozygous TrJ/TrJ genotype leads to the most severe form and mice death before the 17 or 18 days old [22,29,30]. We have demonstrated that the phenotypic changes in TrJ/+ mice are evidenced at an early age by tests that evaluate the neuromuscular aspects, such as the mouse suspension tail test (MSTT) and fixed bar test, 100% coincident with the genotyping identification [31]. However, to our best knowledge, and in addition to the neuromuscular disorders characteristic of CMT, there are no studies that have evaluated whether the TrJ/+ mice display a behavioral profile associated with a central involvement at the brain level. 

In this study, we first evaluated how the mice behaved in two knowns and validated anxiety tests (open field test and elevated plus maze). The open field tests and elevated plus maze are non-invasive tests of anxiety that allow evaluating animal behavior of various species (including mice) under psychological stress associated to novelty and social isolation [32,33,34,35,36,37]. After having determined the behavioral profile of both groups (TrJ/+ and +/+ mice) in the aforementioned tests, and knowing that the hippocampus is an area of the brain associated with anxious-type behavior [38,39,40,41,42], we decided to compare the presence of PMP22 in this area of the brain. Therefore, the aim of this study was to determine both the behavior profiles of TrJ/+ and +/+ mice in anxiety tests and the molecular expression of pmp22 in the hippocampus. The presence of PMP22 aggregate forms has also been quantitatively analyzed in CA3 pyramidal neurons of TrJ/+ and +/+ mice. We discuss the importance of PMP2 and its role in the neuropathological Trembler-J phenotype context.

## 2. Materials and Methods

### 2.1. Animals

TrJ/+ mice were obtained from the B6.D2-Pmp22^Tr-J^/J background (Jackson Laboratories) and the colony was raised at the IIBCE animal house in a controlled environment (12 h dark and 12 h light cycle) and an average temperature of 21 ± 3 °C with free access to food and water. The weaning of the mice was conducted at 21 days of age. At this time, mice were numbered by the method of ear punching [43]. The experimental procedures were approved by the local Ethics Committee (CEUA-IIBCE, MEC, Uruguay, protocol number: 002/05/2016). Animals used in behavioral tests, immunohistochemistry, and ISH were 5-month-old males of each background. For the performance of this work, we have used 57 mice. The method employed to euthanize mice was quick cervical dislocation (AVMA Guidelines for the Euthanasia of Animals: 2020 Edition; https://www.avma.org/sites/default/files/2020-01/2020-Euthanasia-Final-1-17-20.pdf, accessed on 10 March 2021).

### 2.2. Mice Phenotyping

Adult TrJ/+ mice could be easily phenotypically distinguished from +/+ adult mice by the MSTT described by [31]. Briefly, the mice were suspended by the tail, and mice unable to open their hind limbs were classified as TrJ/+ mice.

### 2.3. Behavioral Experiments

Nine TrJ/+ and eleven +/+ mice were tested in open field test, and one week later on, the elevated plus maze test. Behavioral tests were performed in a separate testing room with the same temperature and photoperiod conditions as the breeding room. Animals were transported to the testing room in their home cages and left to acclimatize at least 2 h before testing. Behavioral tests were video recorded for further analysis. After each test, the apparatus was cleaned thoroughly with 70% ethanol and allowed to dry completely between test sessions.

#### 2.3.1. Open Field Test 

The open field apparatus consists of a square Plexiglas cage (35 × 35 × 40 cm) with red walls to minimize outside light and noise. The animals were individually placed in the center of the open field and left to move freely during a 10 min period. Latency time to first movement (s), distance moved (m), time spent in the peripheral zone (s), time spent in the central zone (s) (12 cm × 12 cm), time (s) during which the mice were making locomotion movement (TMLM), and velocity (m/s) were scored by Ethovision XT software 7.0 (Noldus, The Netherlands). Moreover, the number of rearings (mouse reared up on its hind limbs irrespective of whether the animal showed on- or off-wall rearing), groomings (number of times that an animal preened its fur or tail with its mouth or forepaws), defecations (number of fecal boli), and the number of headshakes (the animal shook its head from side to side) were recorded [36]. 

#### 2.3.2. Elevated Plus Maze Test 

The EPM consisted of two open and two closed arms (open arms: 30 cm × 5 cm; closed arms: 30 cm × 5 cm, surrounded by 15 cm high walls). The apparatus was made of wood and elevated 40 cm above the floor. Mice were placed into the central platform (5 cm × 5 cm) of the maze facing a closed arm and allowed to explore the maze for 5 min. Ethological parameters (frequency of grooming, rearing, defecations, head shake, and frequency of entries into open and closed arms and total entries) were scored during the experimental session, as well as total time in open and closed arms [36,44].

### 2.4. Brain Processing to Cryostat and Vibratome Sectioning

The samples were processed as previously described [45]. Briefly, the brains of 14 mice (7 TrJ/+, 7 +/+ mice) were dissected immediately after cervical dislocation, immersed in cold freshly prepared 4% paraformaldehyde (PFA) fixative solution in PHEM buffer (60 mM PIPES, 25 mM HEPES, 10 mM EGTA, 2 mM MgCl_2_, adjusted to pH 7.2–7.4 with KOH pellets) for 1 h at 4 °C in an orbital shaker, and kept in new freshly prepared 4% PFA fixative solution at 4 °C for 24 h. Then, the leftovers of PFA were eliminated by washing the brain in PHEM buffer (6 changes, 5 min each change) at 4 °C in an orbital shaker. 

The brains destined to obtain vibratome sections (4 TrJ/+, 4 +/+ mice), were immediately included in a support prepared with 1.5% gelatin and 12% bovine serum albumin polymerized with 1% glutaraldehyde final concentration, and 60 µm vibratome thick sections were obtained (Leica, VT 1000S). They were then kept in PHEM solution at 4 °C until their use in the analysis of unmasked PMP22 aggregates by confocal microscopy.

The brains destined for obtaining cryosections (3 TrJ/+, 3 +/+, mice) for ulterior pmp22 in situ hybridization (ISH) and subsequent PMP22 immunostaining were cryoprotected by immersion in 30% sucrose-PHEM solution at 4 °C, with agitation, until density matching (when the brains fell to the bottom of the tube, typically 24 h). Then, the brains were infiltrated for 2 h in solutions with increasing concentration of Tissue-Tek O.C.T. (Sakura) and decreasing concentration of 30% sucrose-PHEM solution up to pure Tissue-Tek. Then, the samples were included in a fresh Tissue-Tek block, frozen at −20 °C and cryosectioned in 10-micron-thick sections (SLEE Cryostat, Main Z, Type MEV). The cryosections were kept at −20 °C until ISH immunostaining characterization.

### 2.5. Probe Synthesis for In Situ Hybridization

A fragment corresponding to nucleotides 425 to 538 of *Mus musculus* pmp22 gene (NM_008885) was cloned into pSPT19 plasmid. HindIII or EcoRI linearized purified plasmids were used as a template for a sense or antisense probes synthesis, using SP6 or T7 enzymes, respectively. Digoxigenin label was incorporated into probes using the DIG RNA Labeling Kit (SP6/T7) (11175025910, Roche, Basel, Switzerland) following the manufacturer’s instructions.

### 2.6. In Situ Hybridization

Brain cryosections were permeabilized using 0.1% Triton X-100 in 4× saline-sodium citrate (SSC: 0.6 M NaCl, 0.06 M Na_3_C_6_H_5_O_7_, pH 7.00), for 10 min. ISH was performed as previously described [46]. Briefly, endogenous peroxidase was inactivated by cryosections incubation with 0.03% H_2_O_2_ in 4× SSC buffer for 15 min at room temperature (RT). A prehybridization step was performed by incubating cryosections in a hybridization solution (10% dextran-sulfate, 0.1 mg/mL tRNA, 0.5 mg/mL salmon sperm DNA, and 50% formamide in 4× SSC), for 30 min at 50 °C. Hybridization was performed at 50 °C for 2 h with digoxigenin-labeled antisense pmp22-recognizing probe at 0.5 ng/μL in the hybridization solution. The sense probe in the same conditions was used as a negative control. Washes of increasing stringency at RT were performed as follows: 2 washes with 4× SSC for 10 min each, 2 washes with 2× SSC for 10 min each, 2 washes with 1× SSC for 5 min each, 2 washes with 0.5× SSC for 5 min each, and 2 washes with 0.025× SSC for 5 min each. Samples were fixed post-hybridization with 3% PFA in PHEM buffer for 5 min at RT and then washed with PHEM 3 times for 5 min at RT. Hybridized probes were recognized using an anti-digoxigenin HRP-conjugated antibody (Ref#11207733910, Roche, Taufkirchen, Germany), and the signal was developed using the Tyramide Signal Amplification Kit (TSA™ technology, Z25090 Invitrogen, Massachusetts, USA) following the manufacturer’s instructions.

### 2.7. Immunostaining Post-ISH

Subsequently, the expression of PMP22 and small subunit of neurofilament protein (NF-68) were recognized by indirect fluorescent immunostaining of sections or cells, as previously described [46]. Brain cryosections were incubated with a blocking buffer (BB: 5% normal goat serum, 0.1% bovine serum albumin, 0.1 5 mM glycine in PHEM buffer pH 7.4) for 1 h at 37 °C. Specific antibodies were incubated for 60 min at 37 °C in the incubation buffer (IB: 0.1% bovine seroalbumin, 0.15 mM glycine in PHEM buffer, pH 7.4). Cryosections were washed six times, 5 min each at RT with IB, and then incubated with anti-antibodies for 45 min at 37 °C. Three washes were carried out with IB for 5 min each, at RT, and three additional washes with PHEM at the same conditions were also performed. Finally, samples were mounted with ProLong Gold antifade (P10144, Thermo Fisher, Massachusetts, USA). 

The specific antibodies used were rabbit polyclonal anti PMP22 (Cat# AB5685, RRID: AB_240790, Millipore, Taufkirchen, Germany), work dilution (WD) 1:200 and chicken polyclonal anti NF-68 (Cat# ab72997, RRID: AB_1267598, Abcam, Cambridge, UK) WD 1:800. The anti-antibodies used were goat anti-rabbit IgG (H+L) Alexa Fluor 546 (Cat# A-11035, RRID: AB_143051, Molecular Probes, Eugene, USA) WD 1:1000, goat anti-chicken IgY (H+L) Alexa Fluor 633 (Cat# A-21103, RRID: AB_2535756, Thermo Fisher Scientific, Massachusetts, USA), WD 1:1000. Finally, samples were mounted with ProLong Gold antifade.

### 2.8. Immunohistochemistry of Soluble and Aggregated PMP22 to Fluorescent Signal Quantification

For immunostaining aggregated PMP22, the 60 μm thick vibratome sections (anterior hippocampus) were subjected to an epitope retrieval process, with immersion for 60 s in freshly prepared 70% formic acid. After that, the unmasked sections were quickly washed in water and finally immersed in PHEM buffer, whereas for immunostaining soluble PMP22, the vibratome sections were not pretreated (masked sections). Then, the masked and unmasked brain sections followed the same procedure: they were permeabilized 60 min with 0.1% Triton-X 100-PHEM solution at 37 °C. The brain sections were then incubated with the specific antibody anti PMP22 in IB (Cat# ab61220, RRID: AB_944897, Abcam, Cambridge, UK), WD of 1:100 for 60 min at 37 °C and then for 24 h at 4 °C. After that, brain sections were washed with IB three times for 5 min each at RT. Then, the nonspecific binding epitopes were blocked by incubating the sections with BB at RT for 30 min. This was followed by incubation of the anti-antibody Cy5 or Alexa Fluor 546 (Cat# A10523, RRID: AB_2534032 or A11035, RRID: AB_2534093, respectively, Thermo Fisher Scientific, Massachusetts, USA) WD of 1:1000 and DAPI (Cat# D1306, RRID: AB_2629482, Thermo Fisher Scientific, Massachusetts, USA) in IB at 37 °C, for 45 min in darkness. After that, brain sections were washed with IB two times at 37 °C for 5 min followed by four washes in PHEM at RT for 5 min, and these steps were performed in darkness. The entire procedure was performed under free-floating conditions and agitation. After that, brain sections were carefully placed on slides and mounted with ProLong Gold antifade (P36930, Invitrogen), left to dry for 24 h at RT in darkness before confocal microscopy observation.

### 2.9. Hippocampal Dissection

Three male mice per genotype were used for hippocampal dissection. This was performed according to Olivera et al. (2003), with minor modifications [47]. Briefly, the mouse head was firmly held, and large scissors placed just behind the skull to quickly dislocate it from the spinal cord. After that, the mouse was decapitated, the skin removed, and the foramen magnum left exposed. Then, clean fine scissors were introduced into the foramen magnum and the skull bone cut following the midline until the eyes. There, a transverse incision was made. Bones were broken aside to expose the brain that was then carefully removed and cut through the midline. Under a microscope, each hemi-brain was turned down, the medial structures removed, and the hippocampus quickly dissected with other clean scissors. Then, the remaining cortical tissue was removed, and the clean hippocampus collected in TRIzol^®^. 

### 2.10. RNA Purification and RT—qPCR in Hippocampal Tissue 

RNA isolation from dissected hippocampal tissue was performed with TRIzol^®^ reagent (No. 15596026, Invitrogen, Thermo Fisher Scientific, USA), following the manufacturer’s protocol.

Reverse transcription and quantitative PCR were conducted according to Canclini et al. 2020, with modifications [48]. Purified RNA was reverse transcribed into cDNA using anchored oligo(dT)20 primer (No. 12577011, Invitrogen, Thermo Fisher Scientific, USA) and SuperScript III Reverse Transcriptase (No. 18080093, Invitrogen, Thermo Fisher Scientific, USA). cDNA was used to prepare triplicate reactions for qPCR using SYBR Green Universal Master Mix (No. 4309155, Applied Biosystems, Thermo Fisher Scientific, USA) according to manufacturer’s instructions and run on a CFX96 Touch Real-Time PCR Detection System (Bio-Rad, Hercules, CA, USA) using the following PCR conditions: denaturation for 15 s at 95 °C; annealing and extension for 1 min at 60 °C. The levels for each condition were corrected using β-actin as the housekeeping gene. The following primers were used for qPCR: pmp22 forward: 5′-GAATTCCTGTTCCTGTTCTTCTGCCAGCTC-3′pmp22 reverse: 5′-AAGCTTGTAGATGGCCGCTGCACTCATC-3′β-actin forward: 5′-TATGTTGCCCTAGACTTCGAGC-3′ β-actin reverse: 5′-CAGCTCATAGCTCTTCTCCAGG-3′

### 2.11. Confocal Microscopy

In situ hybridization and subsequent immunohistochemistry experiments were visualized using an Olympus FV-300 confocal microscope equipped with a Plan Apo N 60× oil NA 1.42 lens and 488 nm, 543 nm and 633 nm laser lines. 

Brain PMP22 aggregates immunostaining was visualized using a Zeiss LSM 800 confocal microscope with an air scan module. At the beginning of the confocal season, the specific photomultiplier laser maximal levels were fixed with the negative controls of each sample containing no specific antibodies, using mode levels of saturation, until a few brilliant non-specific signals started to appear. Then, all the images of sections containing specific antibodies were taken at the same conditions, in the same confocal microscopy section. The voltage values of the photomultipliers never exceeded the initial ones set with the control samples, and they were lowered when fluorescence intensity saturation appeared. These procedures ensured equal conditions for fluorescence intensity quantification. The z stacks were obtained in confocal microscopy software as follows: the bottom position was fixed when the image of the object of interest appeared (first image) and the top position was established when the image of the object of interest disappeared (last image). The distance between two continuous planes (z-step distance) was 0.25 microns.

### 2.12. Fluorescent Image Analysis

Confocal images were imported into ImageJ software (version 1.53b, RRID: SCR_003070) for fluorescent intensity analysis following these steps: first, the DAPI and PMP22 channels were separated, and quantification of the total PMP22 intensity was then performed for each plane. 

Randomization and blinding procedures to minimize subjective bias when allocating subjects to experimental groups. The images of each mouse had a code to which the researchers did not have access until the analysis was completed.

### 2.13. Statistical Analysis

The normal distribution of the obtained data was evaluated using the Shapiro–Wilk test. Behavioral and parameters with normal distribution were compared (+/+ vs. TrJ/+) using the unpaired Student’s *t*-test; in all cases, the degree of freedom (df) was 18. Behaviors that did not fit a normal distribution were compared using the Mann–Whitney U-test. Fluorescence intensity data of images were compared using the Mann–Whitney U-test. Data obtained from RT-qPCR were analyzed using Student’s *t*-test. All tests underwent two-tailed analysis and the results were considered significant with an alpha level of 0.05. All graphical and statistical analyses were conducted using GraphPad Prism 8 software (GraphPad Prism, RRID:SCR_002798). In this work, there were no excluded data. All outliers were included.

## 3. Results

### 3.1. Behavior in Anxiety Tests Reveal Phenotypic Differences between TrJ/+ and +/+

Open field test

Latency time to first movement was greater in TrJ/+ than +/+ mice (*p* = 0.001, two-tailed, Mann–Whitney U = 0) (Figure 1A) but there were no significant differences between genotypes in the time spent in the center zone (*p* = 0.503, Mann–Whitney U = 40) (Figure 1B) and in the peripheral zone (*p* = 0.331, Mann–Whitney U = 36) (Figure 1C). TrJ/+ mice showed a decrease in the distance traveled (*p* = 0.001, *t*-Student = 3.870) (Figure 1D), and the time during which the mice were making the locomotion movement (TMLM) (*p* = 0.007, *t*-Student = 3.069) (Figure 1E), and the locomotion velocity (*p* = 0.0004, *t*-Student = 4.375) (Figure 1F), compared to +/+ mice. On the other hand, the frequency of grooming (*p* = 0.0011, *t*-Student = 3.886) and head shakes (*p* = 0.0001, Mann–Whitney U = 0) was greater in TrJ/+ than in +/+ mice. In contrast, the frequency of rearing was lower (*p* = 0.0004, *t*-Student = 4.305) in TrJ/+ than in +/+ mice (Figure 2A), no statistically significant difference was found in the frequency of defecations between TrJ/+ and +/+ mice (*p* = 0.5716, *t*-Student = 5.76) (Figure 2A).

Elevated plus maze test

The frequencies of grooming (*p* = 0.0016, Mann–Whitney U = 10.5), defecations (*p* = 0.0374, Mann–Whitney U = 23.5), and head shake (*p* = 0.0002, Mann–Whitney U = 7) were greater in TrJ/+ than in +/+ mice, while the frequency of rearing was lower in TrJ/+ than in +/+ mice (*p* = 0.0011, *t*-Student = 3.875) (Figure 2B). 

As shown in Figure 3, TrJ/+ mice performed fewer entries to the closed arms (*p* = 0.0124, *t*-Student = 2.780), fewer total entries (*p* = 0.0057, *t*-Student = 3.141), and showed a tendency to present fewer entries in the open arms (*p* = 0.096, Mann–Whitney U = 28) of the maze than +/+ mice (Figure 3). However, no statistically significant difference was found between groups in the percentage of entries in open arms (+/+: 20.0 ± 15.0 s vs. TrJ/+: 0.0 ± 8.3, *p* = 0.2618, Mann–Whitney U = 35) The time spent in open arms (+/+: 25.0 ± 15.3 s vs. TrJ/+: 14.0 ± 13.8 s, median ± SIR; *p* = 0.5480, Mann–Whitney U = 13.5) and closed arms (+/+: 247.0 ± 9.5 s vs. TrJ/+: 278.0 ± 28.3, median ± SIR; *p* = 0.5303, Mann–Whitney U = 13.0) did not show significant differences between groups.

### 3.2. Pmp22 mRNA and PMP22 Are Expressed in Murine Hippocampal CA3 Neurons

The pmp22 transcript and PMP22 protein are expressed in the hippocampal CA3 neurons of both +/+ and TrJ/+ brains (Figure 4A,B). Transcript and protein were detected in the cytoplasm, with the main fluorescent intensity in the perinuclear domains and with less intensity at the nuclear localization. It is remarkable that in CA3 neurons, the pmp22 hybrid and PMP22 protein fluorescence intensity differed between genotypes, being highest in the +/+. However, even though the presence of pmp22 mRNA has been also confirmed in whole isolated hippocampi of both +/+ and TrJ/+ by RT-qPCR, the quantitative analysis showed no significant differences between genotypes (*p* = 0.6646, *t* = 0.4672, df = 4) (Figure 4C).

### 3.3. Soluble and Aggregates PMP22 Are Present in Adult Hippocampal CA3 Neurons

PMP22 protein is present in the adult hippocampal CA3 neurons (Figure 5A). The soluble form of the protein is observed in intact vibratome brain sections (masked sections) without formic pretreatment (Figure 5B). The formic acid pretreatment of coronal brain sections eliminates the soluble form of PMP22 and allows us to recognize the aggregate form of the PMP22 protein (Figure 5C). PMP22 fluorescence intensity shows the highest values in TrJ/+ in both masked and unmasked brain sections, with statistical differences when compared with the same regions of +/+ hippocampus (masked: *p* < 0.0001, Mann–Whitney, U = 314; unmasked: *p* < 0.0001, Mann–Whitney U = 556) (Figure 5B,C).

## 4. Discussion

Anxiety tests, such as the elevated plus maze and open field tests, are non-invasive tests that allow evaluating animal behavior under psychological stress associated with novelty and social isolation [32,49,50]. They allow us to evaluate how animals behave under conditions of psychological stress due to novelty and social isolation. First, their responses involve different elements at the brain level [49,50,51,52]. Secondly, if there are differences in the behavioral profile between both genotypes in the anxiety tests, such changes could indicate brain involvement [38,39,40,41,42]. We explored the expression and localization of pmp22 in some brain areas related to anxiety, such as the hippocampus. We showed that TrJ/+ and +/+ mice present different behavioral profiles in two tests of anxiety and that some changes suggest a central implication at the brain level. In anxiety tests, TrJ/+ mice presented alterations in motor activity, as shown by a lower frequency in the rearing, total distance moved, TMLM, and velocity, compared to +/+ mice. These differences could be related to a peripheral neuropathy similar to CMT in the TrJ/+ mice, to an increased response of anxiety (as a central symptom), or a combination of both. The fact that TrJ/+ (but not +/+) mice presented motor alterations, such as a greater latency, lower TMLM, and lower velocity, could be directly associated with CMT disease, in which the speed of nerve conduction is slower, which are also primarily associated with alterations in myelination of Schwann cells [5,53,54]. In addition, the lower frequency of rearing and the shorter distance moved in OF, and lower number of open and closed arms entries in EPM, could also be associated with peripheral alterations in myelination, which is shown in individuals with CMT1 due to a greater weakness of the hind limbs [5,53], possibly affecting their performance. Similar behaviors of distal (hind limbs) muscle weakness and spastic paresis are observed in the TrJ/+ mouse model [8,25,29,30,31,55,56].

In both anxiety tests, the same profile of three behavioral parameters was observed given that the TrJ/+ mice had a lower frequency of rearing and greater frequency of grooming and head shakes than the +/+ mice. According to other studies, less displacement, less rearing, and more grooming activity were associated with a greater degree of stress or anxiety [57,58,59,60]. Moreover, the greater number the defecations in EPM and the greater latency to the first movement by TrJ/+ mice in OF support the possibility of greater anxiety. Changes in all these behaviors were associated with certain brain areas, particularly the hippocampus. More defecations and greater latency time have been associated with anxiety-type behaviors in mice, being more evident in mice with hippocampal lesions [61]. Moreover, mice with lesions of the hippocampus had a lower frequency of rearing [61,62]. In rodents, the grooming is induced by adrenocorticotropic hormone (ACTH, dose-dependent effect), possibly in response to novelty stress in arena tests as well as for the involvement of certain neurotransmitters, and was also even associated with some brain regions such as the hippocampus [63]. The response of grooming induced by ACTH was diminished by lesions in the hippocampus and substantia nigra [63,64]. Hippocampal lesions in mice decreased the frequency of grooming [61,65] and increased grooming time [61]. The higher frequency of grooming activity in TrJ/+ mice could be a way of runaway oriented to resolve or adapt to conflictive situations of stress or anxiety [63,66,67,68]. Among the neurotransmitters, dopaminergic status is implicated in the manifestation of grooming [69], and this behavior is partly regulated by dopamine D1 receptors [70,71]. All this information, as a whole, suggests that the greater deployment of grooming in TrJ/+ mice could be directly associated with changes at the brain level, in particular, the hippocampus. 

In both tests, TrJ/+ manifested characteristic head shakes, unlike the +/+ mice that essentially did not. This behavior has been reported in some central pathologies, such as serotonin-toxicity syndrome (toxidrome) [72]. In this toxidrome, the 5-HT2A receptors have been implicated in the head shake movement [72,73,74,75,76]. In addition, Hawkins et al. (2008), conducting studies of rodents in the OF and using spiperone (5-HT2A antagonist) and SDZ SER-082 (5-HT2C antagonist), suggested that 5-HT2C receptor, or combined 5-HT2A and 2C receptor, may play an important role in mediating the behavior of head shakes. Therefore, the head shakes observed in the TrJ/+ mice in this work could also be indicative of an alteration at the brain level. Although this type of mouse with the TrJ/+ genotype was named this way, due precisely to the tremor they manifest, we have not found previous works reporting the cause of such tremors. However, there are no publications that address the cause of such a central tremor. It would be interesting to evaluate whether the use of 5-HT2A and 5-HT2C receptor antagonists reverses the head-shaking movement in TrJ/+ mice.

In the present work, we report that some behaviors displayed by the TrJ/+ mice in anxiety tests could be associated with peripheral nerve disorders and, also, with implications of the central level. The differences in the profile of grooming and defecations give us information about brain involvement. Moreover, we report, for the first time, the location of the PMP22 protein in the hippocampus in an animal model (TrJ/+ mice) of the human peripheral neuropathy CMT1E. The differences in molecular expression of pmp22 observed between TrJ/+ and +/+ at the hippocampal level could help to explain, at least in part, the different behavioral profiles associated with each genotype. 

The central expression of pmp22 (transcript and protein) was demonstrated here by ISH in the granular cells of the hippocampus in both TrJ/+ and +/+ mice. It is important to note that the probe used in this work recognizes the coding region of the gene, so all gene transcripts having this region could be recognized by it. The expression of pmp22 transcript seems to show genotypic differences in both distribution and intensity. 

Multiple roles have been described for pmp22/PMP22, mainly in the PNS associated with the compact myelin structure [6,8,77,78,79,80,81,82,83]. Different isoforms of pmp22 and its protein were also signaled in cranial and spinal motor neurons by in situ hybridization and autoradiography [10]. However, the central function of pmp22 and their protein are still uncertain. We can hypothesize that the initially indicated role for pmp22 as a growth arrest-specific gene (gas-3, Manfioletti et al. 1990) [84], described in NIH3T3 fibroblasts, could also be fulfilled in the granular neurons of the hippocampus, associated with the G0 stage maintenance of these cells under physiological conditions. 

The hippocampal presence of the pmp22 transcripts in TrJ/+ and +/+ mice constitutes an important milestone in the confirmation of the central expression of the gene. Moreover, the transcript identified by ISH has been evidenced in TrJ/+ and +/+ hippocampal neurons. Even though the signal cannot be quantified, it seems to be less intense in TrJ/+ than in +/+. However, no statistical differences have been observed between genotypes when hippocampal RT-qPCR analysis have been performed. Pmp22 was initially described in NIH3T3 fibroblasts as a specific cell growth arrest gene (gas-3, Manfioletti et al. 1990) [84]. In this sense, we hypothesized that pmp22 could play a role in growth arrest, maintaining the granular neurons of the hippocampus in the G0 stage of the cell cycle. This role could be adequately accomplished under physiological conditions of +/+ genotype. However, in TrJ/+, point mutation on pmp22 could possibly determine that it could not adequately fulfill its growth arrest function. Furthermore, in hippocampal neurons, the soluble and aggregated PMP22 signals were shown to be significantly different between genotypes, both being higher in the pathological condition of TrJ/+ mice, probably due to imbalance protein processing route. This could imply neuronal dysfunction associated with a PMP22 toxic function gain. The balance between soluble and aggregated forms of PMP22 could determine a specific functional flux, which will need to be more extensively quantified and further characterized in TrJ/+ and +/+, to understand their neural roles. Several experimental approaches support the assumption that PMP22 could possess a dual function: a structural peripheral myelin component and a more complex role related to cell cycle arrest. In this sense, it has been indicated that *pmp22* gene expression was related to reduction of proliferation, maintenance of G0 or cell-differentiated state, and cell death program [13,14,84,85]. Moreover, in Schwann cells held in G0 (non-post-mitotic cells), the presence of PMP22 could incorporate the modulation of other processes. The greater presence of aggregated PMP22 observed in hippocampal neurons in TrJ/+ was previously reported in Schwann cells [46,86]. Fabbretti et al. (1995) [87] proposed PMP22 could participate at the crossroad of alternative cell fates: continued cellular division, growth arrest, differentiation into myelin-forming cells in SC and, also, apoptosis. In myelin deficiency phenotype (Trembler and Trembler-J), SC overexpress PMP22 [88] and continue to proliferate in peripheral nerve [89], acquiring a profile of SC development markers that resembles the characteristics of normal SC before myelination [90].

In future works, it would be interesting to explore the presence and the role of nuclear PMP22, even in both hippocampal neurons and SC and its relation with p53 in the balance between these two pathways through the alternative expression of PMP22 and p53-apoptosis-effector-related-to-PMP22 (PERP), as proposed by Attardi et al. (2000) [91]. The balance between soluble and aggregated forms of PMP22 could determine a specific functional flow that must be more extensively quantified and characterized in +/+ and TrJ/+. This topic should also be explored in the future to begin to elucidate the role of pmp22 in neuronal and glial cells nucleus. 

As we pointed out before, the pmp22 transcript was early evidenced in the brain and spinal cord in adult and embryonic rats and mice [10,92], and much later, in the human brain [17]. The PMP22 protein was also identified in myelin isolated from adult human and mouse brains [10,77]. Pyramidal signs were described by Thomas et al. (1997) [93] in three CMT1A (triploidy of pmp22) patients and Chanson et al. (2013) [94], reported a reduction of white matter volume and cognitive impairment in 70% of CMT1A studied patients. More recently, Brandt et al. (2016) [95] have also described functional, metabolic, and macrostructural alterations in the afferent visual system in eighteen patients with HNPP (pmp22 haploidy). Taking together these previous findings, they further provide some understanding related to the general role of pmp22 expression, especially in the CNS, as well as to normal gene copy number (genic dosage effect). In this context, our results contribute to the understanding of the physiological role of pmp22 expression in hippocampal neurons, while the pmp22 mutation could explain, at least in part, behavioral differences associated with it.

Finally, our study evidenced the presence of pmp22/PMP22 at the brain level, more specifically in the hippocampus in +/+ and TrJ/+ mice. Therefore, this study opens new perspectives on the physiology of pmp22 in the nervous system.

## 5. Conclusions

In conclusion, we demonstrate that TrJ/+ mice (carrying a mutation in the pmp22), as an animal model of CMT1E disease, have a different behavioral profile than +/+ mice in anxiety tests, evidencing the involvement of the CNS. The presence of aggregated PMP22 in granular hippocampal neurons was higher in TrJ/+ than in +/+ mice. Furthermore, we showed, for the first time, the presence of pmp22 transcripts and PMP22 protein in the hippocampal domain of brain sections of +/+ and TrJ/+ mice. In conclusion, the data gathered in this work reveal that TrJ/+ mice, in addition to the peripheral manifestations of the pathology, present a clear involvement of the CNS. Further studies are needed to more deeply unravel the potential link between differences found in the behavioral profile and peculiar expression of pmp22/PMP22 in Trj/+ compared to +/+ male mice.

## Figures and Tables

**Figure 1 biomolecules-11-00601-f001:**
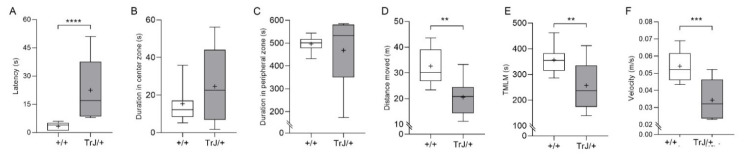
Movement parameters of +/+ mice and TrJ/+ mice in the open field test. (**A**) Latency. (**B**) Duration in the center zone. (**C**) Duration in the peripheral zone. (**D**) Distance moved. (**E**) TMLM: time during which the mice were making locomotion movement. (**F**) Velocity. Parameters in (**A**–**C**) were not parametrically distributed and they were analyzed using the Mann–Whitney U-test. Parameters in (**D**–**F**) were normally distributed and analyzed using Student’s *t*-test, df = 18. Asterisk indicates a significant difference between TrJ/+ (gray bars, *n* = 9) and +/+ (white bars, *n* = 11). ** *p* < 0.01, *** *p* < 0.001, **** *p* < 0.0001. The mean is shown as “+”.

**Figure 2 biomolecules-11-00601-f002:**
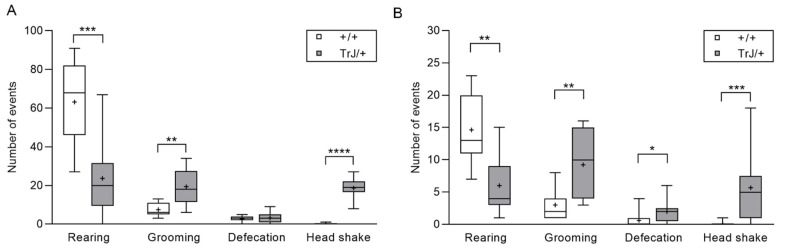
Rearing, grooming, defecations, and head shake of +/+ mice and TrJ/+ mice. (**A**) Parameters analyzed in open field test. (**B**) Parameters analyzed in elevated plus maze test. Rearing and grooming behaviors were analyzed using Student’s *t*-test, df = 18. Defecations and head shake were analyzed using the Mann–Whitney U-test. Asterisk indicates significant difference between TrJ/+ (gray bars, *n* = 9) and +/+ (white bars, *n* = 11) mice: * *p* < 0.05, ** *p* < 0.01, *** *p* < 0.001, **** *p* < 0.0001. The mean is shown as “+”.

**Figure 3 biomolecules-11-00601-f003:**
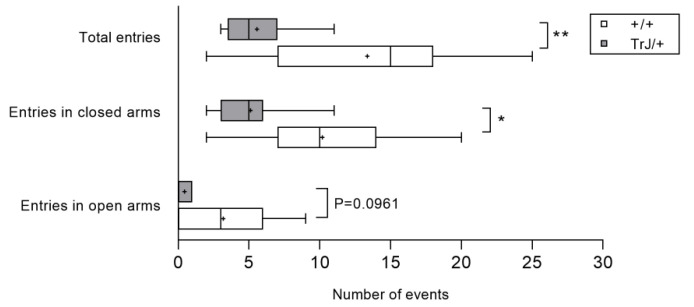
Entries in open and closed arms of +/+ mice and TrJ/+ mice in the elevated plus maze test. The behavioral parameters total and closed arms were analyzed using Student’s *t*-test, and for open arms, was analyzed using the Mann–Whitney U-test. Asterisk indicates a significant difference between TrJ/+ (gray bars, *n* = 9), and +/+ (white bars, *n* = 11) mice: * *p* < 0.05, ** *p* < 0.01. The mean is shown as “+”.

**Figure 4 biomolecules-11-00601-f004:**
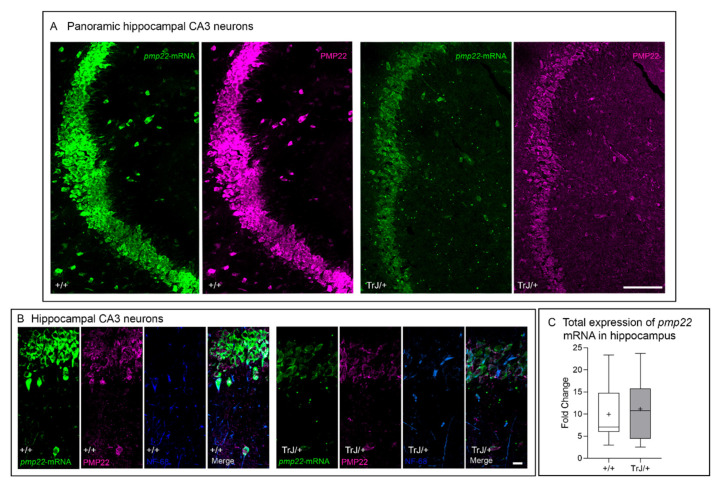
In situ hybridization of pmp22 mRNA and associated PMP22 immunostaining in adult hippocampal CA3 neurons. (**A**) A panoramic view of the CA3 hippocampal regions show the ISH pmp22 signal (green) and their associated PMP22 (soluble form of protein) (magenta) in +/+ and TrJ/+ genotypes. (**B**) In situ hybridization of pmp22 mRNA in hippocampal adult CA3 neurons of +/+ and TrJ/+ mice brains, showing the pmp22 transcript was detected in +/+ and TrJ/+ brain cryosections at the hippocampal CA3 region. The hybrid signals were observed in nuclear and perinuclear domains. Moreover, correlative post-ISH immunostaining of PMP22 and NF-68 may be observed. The upper panel shows +/+ and the lower panel shows the TrJ/+ fluorescence intensity of hybrid and proteins. (**C**) Hippocampal pmp22 mRNA expression in +/+ y TrJ/+ mice was determined by RT-qPCR. Comparative analysis between +/+ and TrJ/+ hippocampi shows no significant differences (Student’s *t*-test, *p* = 0.62, *n* = 3 for each genotype). mRNA levels were normalized against β-actin mRNA. “+”: mean. The mean is shown as “+”. Scale bar for A = 50 µm for all panels. Scale bar for B = 5 μm for all panels.

**Figure 5 biomolecules-11-00601-f005:**
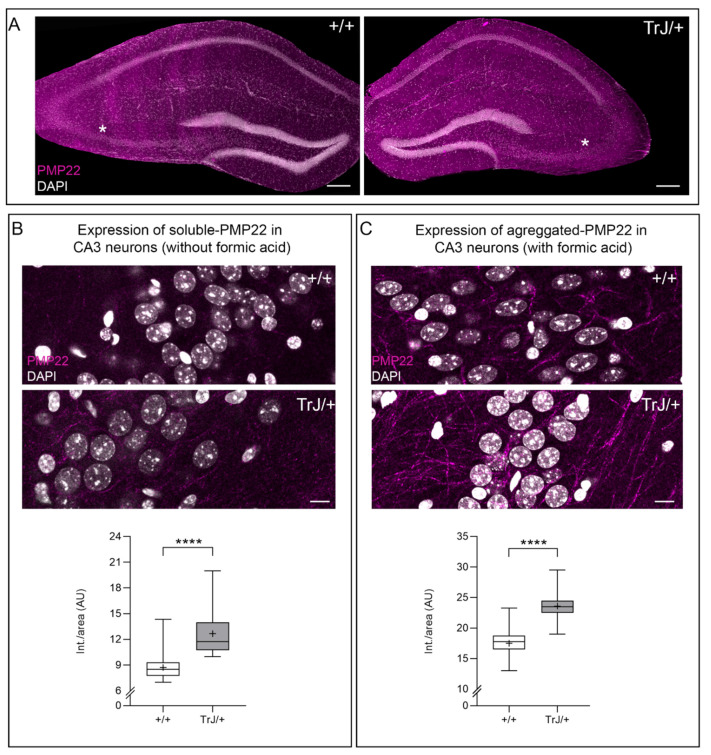
Localization of aggregated and soluble PMP22 in the hippocampus of adult mice. (**A**) A tail scan image of the whole hippocampal formation of both +/+ and TrJ/+ genotypes. The asterisk (*) indicates hippocampal CA3 neurons regions. Nuclear domains were enlightened by specific DAPI staining. Expression of PMP22 (magenta) aggregates in the hippocampus after epitope retrieval (70% formic acid), in 60 μm thick vibratome sections of both +/+ and TrJ/+. (**B**) PMP22 aggregated is shown in the CA3 pyramidal neurons of both +/+ and TrJ/+ mice. (**C**) Specific PMP22 showed significant differences between genotypes, with the highest values observed in TrJ/+ mice, analyzed by the Mann–Whitney U-test (*p* < 0.0001, *n* = 4 for each genotype). Scale bar for A = 200 μm for all panels. Scale for B = 10 μm for all panels. **** *p* < 0.0001. The mean is shown as “+”.

## Data Availability

Data available on request due to restrictions e.g., privacy or ethical. The data presented in this study are available on request from the corresponding author.

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
