# Peer review of "Central Alteration in Peripheral Neuropathy of Trembler-J Mice: Hippocampal pmp22 Expression and Behavioral Profile in Anxiety Tests"

_biomolecules, 2021, doi:10.3390/biom11040601_

Round 1
Reviewer 1 Report
The authors used a mice model to study the Charcot-Marie-Tooth (CMT) type 1 disease by behavioral tests and molecular experiments. There were many critical details missing in the introduction and methods, which limited the the significance and validity of this research.
Major comments:
- Could the authors give the age of the mice when behavioral tests were performed? Furthermore, what's age of pathogenesis of CMT1E in human patients. Could the mice model mimic the development of CMT1E in human?
- Does CMT1E have gender difference in human? What gender of mice were used in this study? Is it possible that gender could influence the behavioral results?
- There are many brain regions that highly related to anxiety, such as amygdala, raphe nucleus, BNST and hypothalamus. Could the authors explain why they only choose hippocampus for the analysis of pmp22 expression?
- Hippocampus is well-known for adult neurogenesis. Could pmp22 have any effect for this process? Why the authors only analyzed CA3 region?
- For elevated-plus-maze test, could the authors calculate the total time spent in open arm and close arm?
Author Response
- A) Authors: The authors thank the reviewers for comments and suggestions that helped improve the manuscript.
(R) Review 1
R: Could the authors give the age of the mice when behavioral tests were performed?
A: When the behavioral tests were carried out, the mice were 5 months old. This information can be found in the first version of the manuscript on sheet 3 line 118 and 119 ("5 month old male mice").
R: Furthermore, what's age of pathogenesis of CMT1E in human patients?
A: The clinical onset of the CMTs is usually within the first two decades of life, they are progressive and the neurodegeneration, affects the quality of life of patients with a variable severity (Vallat, 2003). CMT disease is a human syndrome, by grouping the most frequente hereditary (genetic) neuropathies (prevalence of 1:2500)(Morena et al, 2019). Within them, demyelinating neuropathies (CMT1) have mutations that alter the structural integrity of compact and non-compact myelin. Mutations affecting the pmp22 gene, which codes for the compact myelin protein PMP22, account for approximately 70% of myelinopathies (Vallat, 2003). Within CMT1, CMT1E is a group of myelinopathies caused by different point mutations in the pmp22 gene, that represent the 2,5% of all CMTs. Nevertheless, the clinic shows a very large range of severity from mild (similar to HNPP), to very severe demyelinating neuropathies, like CMT1-DSS syndrome (Russo et al, 2011, Meyer zu Hörste & Nave, 2006). The known mutations of PMP22 inside CMT1E, include 44 single base substitutions, 14 deletions, 2 insertions, one reciprocal translocation, several excision sites and some single base substitutions in exon 1A and in the 3' UTR. With a few exceptions, most of the mutations observed in PMP22 show autosomal dominant heritability (Russo et al, 2011; Suter & Scherer, 2003). The evaluation of CMT1E pathogenesis has become an interesting topic, as it has revealed important mechanisms of PMP22 function. The use of two animal models of CMT1E, the Trembler and Trembler-J mice, has elucidated some of the heterogeneity of the CMT1E clinical phenotype, caused by intracellular trafficking and defective processing of mutated PMP22 in Schwann cells, affecting the whole nerve fiber (Li et al 2013; Russo et al, 2011).
References:
Vallat, J.-M. Dominantly inherited peripheral neuropathies. J. Neuropathol. Exp. Neurol. 62 , 699–714 (2003).
Suter, U. & Scherer, S. S. Disease mechanisms in inherited neuropathies. Nat. Rev. Neurosci. 4 , 714–726 (2003)
.
Russo, M. et al. Variable phenotypes are associated with PMP22 missense mutations. Neuromuscul. Disord. 21 , 106–114 (2011).
Li, J., Parker, B., Martyn, C., Natarajan, C. & Guo, J. The PMP22 gene and its related diseases. Mol.Neurobiol. 47 , 673–698 (2013).
Morena J, Gupta A, Hoyle M Int J Mol Sci. 2019 Jul 12;20(14):3419. doi: 10.3390/ijms20143419.
Meyer Zu Hörste G, Nave KA. Animal Models Curr Opin Neurol. 2006 Oct;19(5):464-73. doi: 10.1097/01.wco.0000245369.44199.27.
R: Could the mice model mimic the development of CMT1E in human?
A: This is mentioned on page 2 from line 80 to 88 of the introduction. The Trembler J model shares a spontaneous mutation in the pmp22 gene with a human family (Valentijn et al, 1992). This confers to TrJ/+ model a high genetic “fidelity” with the disease, constituting a valuable tool for the study of CMT1E. As it is an inherited condition, the model allows its development to be studied at any stage of the disease. In Trembler-J the severity of the disease is gene dose-dependent, as only heterozygotes (TrJ/+) are viable (recessive homozygotes die before weaning). Clinically, TrJ/+ mice manifest spastic paralysis and generalised tremor. The mutation is located on mouse chromosome 11, which has regions of conserved synteny with human chromosome 17 (Buchberg, et al, 1989). The mutation (L16P), affects the first transmembrane domain of the final protein, (Burchberg, et al 1989; Suter et al 1992; Suter, U. et al., 1992; Notterpek, et al., 1997), which prevents the normal folding of the protein and modifies its insertion into the membrane, causing it to accumulate inside the cell. These proteins not only lose the ability to fold correctly, but also acquire a toxic effect for the Schwann cell (Quarles, et al.,2006; Tobler, et al.,2002). The presence of cytoplasmic aggregates of mutated protein forming heterodimers alone or with the normal protein, have been described (Fortum et al 2007; Notterpek & Tolwani, 1999). TrJ/+ pheripheral nerves show markedly decreased myelination, with axonal degeneration profiles, Schwann cell proliferation and a redundant basal lamina (Myers et al, 2008). Several evidences point to Trembler-J mice as a valid model to study hereditary peripheral neuropathies (Fortum et al 2005; Henry et al,1983), because they mimicking pathological features of patients born with the same mutations (Tobler et al, 2002, Valentijn et al 1992).
References:
Fortun, J. et al. Impaired proteasome activity and accumulation of ubiquitinated substrates in a hereditary neuropathy model. J. Neurochem. 92 , 1531–1541 (2005).
Buchberg, A. M., Brownell, E., Nagata, S., Jenkins, N. A. & Copeland, N. G. A Comprehensive Genetic Map of Murine Chromosome 11 Reveals Extensive Linkage Conservation Between Mouse and Human . Genetics Soc America (1989).
Myers, J. K., Mobley, C. K. & Sanders, C. R. The peripheral neuropathy-linked Trembler and Trembler-J mutant forms of peripheral myelin protein 22 are folding-destabilized. Biochemistry 47 ,10620–10629 (2008).
Suter, U. et al. Trembler mouse carries a point mutation in a myelin gene. Nature 356 , 241–244 (1992).
Suter, U. et al. A leucine-to-proline mutation in the putative first transmembrane domain of the 22-kDa peripheral myelin protein in the trembler-J mouse. Proc. Natl. Acad. Sci. U. S. A. 89 , 4382–4386 (1992).
Notterpek, L., Shooter, E. M. & Snipes, G. J. Upregulation of the endosomal-lysosomal pathway in the Trembler-J neuropathy. J. Neurosci. 17 , 4190–4200 (1997).
Quarles Richard H., Macklin Wendy B. & Morell Pierre. PART I Cellular Neurochemistry and Neural Membranes, CHAPTER 4 Myelin Formation, Structure and Biochemistry . books.google.com (2006). doi:American society for neurochemistry
Tobler, A., Liu, N., Mueller, L. & Shooter, E. Differential Aggregation Of The Trembler And Trembler J Mutants Of Peripheral Myelin Protein 22. J. Peripher. Nerv. Syst. 7 , 206–207 (2002).
Henry, E. W., Cowen, J. S. & Sidman, R. L. Comparison of trembler and trembler-j mouse phenotypes: Varying severity of peripheral hypomyelination. J. Neuropathol. Exp. Neurol. 42 , 688–706 (1983).
Fortun, J. et al. The formation of peripheral myelin protein 22 aggregates is hindered by the enhancement of autophagy and expression of cytoplasmic chaperones. Neurobiol. Dis. 25 , 252–265 (2007).
Notterpek, L. & Tolwani, R. J. Experimental models of peripheral neuropathies. Laboratory Animal Science 49 , 588–599 (1999).
Valentijn, L. J. et al. Identical point mutations of PMP-22 in Trembler-J mouse and
Charcot-Marie-Tooth disease type 1A. Nat. Genet. 2 , 288–291 (1992).
R: Does CMT1E have gender differences in human?
A: To our knowledge, there are no studies linking specific CMT1E disease manifestations to a particular gender. Interestingly, PMP22 has been described in association with different integrins in endometrial tissue and in a human endometrial adenocarcinoma cell line (HEC-1 A) (Rao 2011). However, the implication of these roles in women with CMT1E, nor in Trembler-J females, has not been specifically explored so far.
R: What gender of mice were used in this study?
A: The animals used in this work are all males, as described in the first version of the manuscript, on page 3 line 119.
R: Is it possible that gender could influence the behavioral results?
A: The reviewer's question is interesting. However, the objective of our work was not to compare the response in anxiety tests in different genders. In this sense, we do not know if male mice could respond differently than a female according to whether they present CMT1E or not. On the other hand, it is known that the response to anxiety tests varies according to sex in mice (Jefferson et al., 2020; Ribeiro-Carvallo et al., 2020; Oliveira de Matos et al., 2020). In addition, within the female sex, the behavioral response in anxiety tests in mice also varies according to the state in the sexual cycle (Jaric et al., 2019; Datta et al., 2019). Testing animals according to sex and the state of the sexual or reproductive cycle could be interesting to evaluate in future works. But besides that, it is not part of the objective of this study (evaluate the influence of sex), it is important to consider that would imply greatly increasing the number of animals to use, which could also have implications from the point of view of animal welfare (Baumans, 2005; Lindsjö et al., 2016).
References:
Baumans V. Science-based assessment of animal welfare: laboratory animals. Rev Sci Tech. 2005 24(2):503-13.
Datta S, Samanta D, Tiwary B, Chaudhuri AG, Chakrabarti N. Sex and estrous cycle dependent changes in locomotor activity, anxiety and memory performance in aged mice after exposure of light at night. Behav Brain Res. 2019 365:198-209.
de Matos LO, de Araujo Lima Reis AL, Lopes Guerra LT, de Oliveira Guarnieri L, Moraes MA, Arabe LB, de Souza RP, Pereira GS, Souza BR. Early postnatal l-Dopa treatment causes behavioral alterations in female vs. male young adult Swiss mice. Neuropharmacology. 2020 170:108047
Jaric I, Rocks D, Greally JM, Suzuki M, Kundakovic M. Chromatin organization in the female mouse brain fluctuates across the oestrous cycle. Nat Commun. 2019 10(1):2851.
Jefferson, S.J., Feng, M., Chon, U.R., (...), Kim, Y., Luscher, B. Disinhibition of somatostatin interneurons confers resilience to stress in male but not female mice: GABAergic control of resilience to stress. 2020 Neurobiology of Stress 13,100238
Lindsjö J, Fahlman Å, Törnqvist E. Animal welfare from mouse to moose--implementing the principles of the 3rs in wildlife research. J Wildl Dis. 2016 52(2 Suppl):S65-77.
Ribeiro-Carvalho A, Lima CS, Dutra-Tavares AC, Nunes F, Nunes-Freitas AL, Filgueiras CC, Manhães AC, Meyer A, Abreu-Villaça Y. Mood-related behavioral and neurochemical alterations in mice exposed to low chlorpyrifos levels during the brain growth spurt. PLoS One. 2020 15(10):e0239017.
R: There are many brain regions that highly related to anxiety, such as amygdala, raphe nucleus, BNST and hypothalamus. Could the authors explain why they only choose hippocampus for the analysis of pmp22 expression?
A: We agree with the reviewer's comment that there are many brain regions linked to the anxiety response. The objective of this study was centered on one of them, and for which we chose the hippocampus. We chose to evaluate the hippocampus given that it is a region clearly involved in the anxiety response, and some of these elements were mentioned in the discussion of this manuscript, see page 12 line 493-508: “Changes in all these behaviors were associated with certain brain areas, particularly the hippocampus. More defecations and greater latency time have been associated with anxiety-type behaviors in mice, being more evident in mice with hippocampal lesions [61]. Also, mice with lesions of the hippocampus had a lower frequency of rearing [61,62]. In rodents, the grooming is induced by Adrenocorticotropic Hormone (ACTH, dose-dependent effect), possibly in response to novelty stress in arena tests, as well as for involvement of certain neurotransmitters, even also was associated with some brain regions such as the hippocampus [63]. The response of grooming induced by ACTH was diminished by lesions in the hippocampus and substantia nigra [63,64]. Hippocampal lesions in mice decreased the frequency of grooming [61,65] increased grooming time [61]. The higher frequency of grooming activity in TrJ/+ mice could be a way of runaway oriented to resolve or adapt to conflictive situations of stress or anxiety [63,66–68]. Among neurotransmitters, dopaminergic status is implicated in the manifestation of grooming [69], and since this behavior is partly regulated by dopamine D1 receptors [70,71]. All this information suggests that the greater deployment of grooming in TrJ/+ mice could be directly associated with changes in a brain level, particularly in the hippocampus.”
If the reviewer considers necessary to add more information about the hippocampus and the anxiety response in mices we could do so.
According to the comments of the reviewer, and from the first findings shown in this work, a new window of future studies opens so that we can evaluate other regions of the brain and combine it with other behavioral tests.
R: Hippocampus is well-known for adult neurogenesis. Could pmp22 have any effect for this process?
A: The question raised by the referee is of the most importance. To our knowledge, the involvement of pmp22 at the central level has not been reported in the hippocampal region, and our work is the first to explore it. We have found no literature associated with neurogenesis in TremblerJ. We have begun studies to characterise possible neurogenesis in these mice and its association with pmp22 expression. Our preliminary results appear to show differences in pmp22 expression in the dentate gyrus of TrJ and wt but associated studies involving neurogenesis measured by BrDU need to be completed. Further refinement of the hippocampal study, extending it to other regions of the Horn of Ammon and the inclusion of the dentate gyrus, is part of our future goals.
R: Why the authors only analyzed CA3 region?
A: In our work we have chosen CA3 due to its significant roles in memory and higher susceptibility to seizures and neurodegeneration (Cherubini and Miles 2015). CA3 pyramidal extensively contact with neighbor excitatory and inhibitory neurons forming complex and or recurrent circuits that are implicated in coding of spatial representations and episodic memories and also receive inputs from the entorhinal cortex that are essential for memory formation (Deuker et al. 2013; Cherubini and Miles 2015). CA3 generates coherent population synchronies that appear to associate firing in selected groups of cells in different behavioral conditions (Buzsáki et al., 1983). In addition, abnormalities in migration of CA3 neurons resulted in defective lamination that is implicated in multiple inherited neurological and psychiatric disorders (Belvindrah et al. 2014). Furthermore, as Belvindrah et al. (2014) propose “the specific CA3 molecular and cellular features, including more complex curved routes and longer pausing steps during cell migration, as well as a reduced potential of genetic and functional redundancy, that may all contribute to a higher vulnerability of this region”.
References:
Belvindrah R, Nosten-Bertrand M, Francis F (2014) Neuronal migration and its disorders affecting the CA3 region. Front. Cell. Neurosci. 8:63. 10.3389/fncel.2014.00063
Buzsáki G, Leung LW, Vanderwolf CH (1983) Cellular bases of hippocampal EEG in the behaving rat. Brain Res. 287, 139–171.
Cherubini E, Miles R (2015) The CA3 region of the hippocampus: how is it? What is it for? How does it do it? Front Cell Neurosci. 9:19. doi: 10.3389/fncel.2015.00019. PMID: 25698930; PMCID: PMC4318343.
Deuker L, Olligs J, Fell J, Kranz TA, Mormann F, Montag C et al (2013) Memory consolidation by replay of stimulus-specific neural activity. J. Neurosci. 33, 19373–19383.
R: For elevated-plus-maze test, could the authors calculate the total time spent in open arm and close arm?
A: Responding to Reviewer we include the total time spent in open arm and close arm. This information is now in the manuscript in page 4 lines 168 and 169 and page 8 line 383 to 386.

Reviewer 2 Report
Biomolecules
Manuscript ID: biomolecules-1159619
Central alteration in Peripheral Neuropathy of Trembler-J mice: hippocampal pmp22 expression and behavioral profile in anxiety tests
The authors determined expression and functions of PMP22, whose mutations are the most common cause of Charcot-Marie-Tooth (CMT) type 1 disease, in the CNS.
The Trembler-J mice (TrJ/+) showed increased anxiety-related behaviors that are considered to be associated with hippocampal functions.
They also showed pmp22 mRNA and protein expression in the hippocampus and cultured hippocampal neurons by in situ hybridization and immunostaining methods, respectively.
The adult TrJ/+ mice showed aggregated form of PMP22 in the hippocampus, which can be attributed to incorrect folding of PMP22.
This study provides important novel findings on pmp22 expression and functions in the CNS and the manuscript is written clearly and precisely.
There are, however, several concerns that should be addressed before publication.
Major comments,
- The expression of PMP22 in the CNS is an important issue of the study. The authors need to show it directly by western blotting.
- In Figure 6, normal expression of PMP22 in hippocampal tissue should also be shown without the formic acid treatment although the presence of PMP22 aggregates forms in the adult mice is informative. Please use makers for neuron and oligodendrocyte with PMP22 antibody because PMP22 is known to be detected in myelin in both the PNS and CNS.
- In Figure 4, cultured neurons do not look healthy. Why dose NF-68 show punctiform in the cell body, without neurite expression, in WT neuron? Please provide better images of cellular expressions of pmp22 mRNA and protein.
Author Response
Reviewer 2:
R: The expression of PMP22 in the CNS is an important issue of the study. The authors need to show it directly by westernblotting.
A: The molecular approach, proposed by the reviewer, is not straightforward and generally provides highly variable results of success for the characterisation of the PMP22 protein with commercially available antibodies. However, in the PMP22 literature, an antibody that does not appear to be commercially distributed but was manufactured by Snipe in 1992 and has been used in a number of papers linked to his research group, stands out for the quality of the WB signals (Spnipes et al, 1992; Paarek et al 1992; Notterpek et al., 1997; Pareek et al 1997, Amicci et al., 2006; Lee et al., 2014 ). To our knowledge, there are no better results in the characterisation by WB of PMP22, nor has any other specific antibody been reported with the quality of the one produced by Dr. Snipes.
In our experience in the biochemical recognition of PMP22, we have performed WB in the past with different commercial antibodies that recognise different domains of the protein (AB5685 Millipore-Chemicom rabbit polyclonal against synthethic peptide of C' terminal of human PMP22; ABCAM ab 61220 rabbit polyclonal against synthethic peptide of C' terminal of human PMP22; SC-65739 mouse monoclonal against 13-mer peptide in the 2nd extracellular domain Santa Cruz Biotechnology), but the results were generally fair to poor. Currently we only have the last two above mentioned antibodies, of which the abcam one is now discontinued, which would not allow repeatability in the future, and the Santa Cruz one has very poor quality.
The structural characteristics of the protein (small size and large regions of hydrophobicity) add an additional difficulty to its solubilisation and often compromise the final obtaining of sufficient amounts of protein for WB analysis.
Finally, due to the conditions of restricted mobility as a preventive measure against the SARS CoV-2 pandemic, we currently have extremely limited access to our laboratories, so that implementing alternative techniques such as immunoprecipitation of PMP22 prior to WB to increase the efficiency of protein recovery in the hippocampus, is unfeasible in the current situation in Uruguay and will remain so for several months to come. If the reviewer has an antibody against PMP22 that works well and is available, we would be happy to test it on our samples.
References:
Sooyeon Lee,1 Stephanie Amici,1 Hagai Tavori,2 Waylon M. Zeng,1 Steven Freeland,1 Sergio Fazio,2and Lucia
Notterpek. 2014 PMP22 Is Critical for Actin-Mediated Cellular Functions and
for Establishing Lipid Rafts. The Journal of Neuroscience, November 26, 2014 • 34(48):16140 –16152
Amici SA, Dunn WA Jr, Murphy AJ, Adams NC, Gale NW, Valenzuela DM, Yancopoulos GD, Notterpek L (2006) Peripheral myelin protein 22 is in complex with _6_4 integrin, and its absence alters the Schwann cell basal lamina. J Neurosci 26:1179 –1189. CrossRef Medline
Pareek S, Notterpek L, Snipes GJ, Naef R, Sossin W, Laliberte´ J, Iacampo S, Suter U, Shooter EM, Murphy RA (1997) Neurons promote the translocation of peripheral myelin protein 22 into myelin. J Neurosci 17:7754– 7762. Medline
Notterpek L, Shooter EM, Snipes GJ (1997) Upregulation of the endosomal-lysosomal pathway in the Trembler-J neuropathy. J Neurosci 17:4190–4200.
Pareek S, Suter U, Snipes GJ, Welcher AA, Shooter EM, Murphy RA (1993) Detection and processing of peripheral myelin protein PMP22 in cultured Schwann cells. J Biol Chem 268:10372–10379.
G J Snipes 1, U Suter, A A Welcher, E M Shooter 1992Characterization of a nágina ovel peripheral nervous system myelin protein (PMP-22/SR13) J Cell Biol.;117(1):225-38.doi: 10.1083/jcb.117.1.225.
R: In Figure 6, normal expression of PMP22 in hippocampal tissue should also be shown without the formic acid treatment although the presence of PMP22 aggregates forms in the adult mice is informative. Please use makers for neuron and oligodendrocyte with PMP22 antibody because PMP22 is known to be detected in myelin in both the PNS and CNS.
A: We thank reviewer 2 for his very valuable suggestion. In response to his request, we have carried out the study of soluble PMP22 expression in the population of hippocampal neurons in the CA3 region, in both genotypes. This was performed following the same protocol applied previously but without applying formic acid pretreatment. Quantification and statistical processing was performed following the same procedure as with the formic acid treated samples. Materials and methods: Page 6, lines 262 to 282; Results: page 11, Fig. 6 and page 12, lines 454 to 462; Discussion: Page 14, lines 555 to 574.
R: In Figure 4, cultured neurons do not look healthy. Why doseNF-68 show punctiform in the cell body, without neuriteexpression, in WT neuron? Please provide better images ofcellular expressions of pmp22 mRNA and protein.
A: We have looked at all our images and they all look similar. We used very low density cultures and the images were taken at high magnification, which is one of the reasons why, at least in part, you see the neurofilament labelling stippling in some cells. We recognise that NF-68 is not the best marker and in our next experiments we will use instead of NF-68 the MAP2 marker, which is more appropriate for this type of culture.
If the reviewer considers that this explanation is not sufficient, we are willing to delete the image and the content corresponding to the embryonic culture of hippocampal neurons, so that the rest of the work can be communicated.

Reviewer 3 Report
The manuscript determines the role of hippocampal pmp22 in peripheral neuropathy of Trembler-J mice. The authors found that behavior profile traits were associated with anxiety and a differential expression of pmp22/PMP22 in hippocampal neurons of TrJ/+ and +/+ mice. According to the author, TrJ/+ mice have a different behavioral profile than +/+ mice in anxiety tests, evidencing the involvement of the CNS. The presence of aggregated-PMP22 in granular hippocampal neurons was higher in TrJ/+ than in +/+ mice. The author presents a nice piece of work that will help in the understanding of the physiological role of pmp22 expression in hippocampal neurons.
- The title and abstract reflect the content of the work.
- The introduction section requires some improvement. Please add the role of pmp22 in other neurological diseases.
- The authors have performed good experimental work.
- Results and discussion were correctly explained.
Author Response
Reviewer 3:
R: The introduction section requires some improvement. Please addt he role of pmp22 in other neurological diseases.
A: The search, based on reviewer 3's question, oriented towards the role of pmp22 in other neurological diseases, has revealed the scarcity of information in the literature. Although there are case report papers (Koros et al. 2013; Wu et al. 1997), where they connect multiple sclerosis with CMT1A, they do not comprehensively delve into the functionality of pmp22 and its implication in multiple sclerosis.
References:
Koros, Christos, Maria-Eleftheria Evangelopoulos, Costas Kilidireas, and Elisabeth Andreadou. 2013. “Central Nervous System Demyelination in a Charcot-Marie-Tooth Type 1A Patient.” Case Reports in Neurological Medicine 2013(Table 1):1–4. doi: 10.1155/2013/243652.
Wu, Aimin;, Lyn; March, Xuanqi; Zheng, Jinfeng; Huang, Xiangyang; Wang, Jie; Zhao, Fiona; M.Blyth, Emma; Smith, Rachelle; Buchbinder, and Damian; Hoy. 1997. Mulúple Sclerosis Associated with Duplicated CMT1A: A Report of Two Cases. Vol. 63.
Round 2
Reviewer 1 Report
The authors clearly explained my concerns.
Author Response
Thank you so much for your requests.
Cordially,
Alejandra Kun
Reviewer 2 Report
Biomolecules
Manuscript ID: biomolecules-1159619
Central alteration in Peripheral Neuropathy of Trembler-J mice: hippocampal pmp22 expression and behavioral profile in anxiety tests
The authors appropriately answered to my comments.
I understood the current situation of the availability of good antibodies for PMP22.
I recommend to delete the images and the results using cultured hippocampal neurons (Figure 4) without staining with other neural markers as the authors mentioned in their responses. These images are not acceptable to cell biologists due to the reason I commented previously.
Author Response
Dear Reviewer:
We have deleted in the text everything suggested by you, concerning the embryonic culture of hippocampal neurons:
1) Materials and methods section: from line 124 to line 133 and from line 171 to line 190.
2) in section Results: from line 403 to line 423.
3) and in the Discussion section: line 574.
In addition, we have deleted figure 4 and its legend.
Thank you for your contribution to the improvement of our work.
Cordially yours,
Alejandra Kun